# Specificity in Ubiquitination Triggered by Virus Infection

**DOI:** 10.3390/ijms21114088

**Published:** 2020-06-08

**Authors:** Haidong Gu, Behdokht Jan Fada

**Affiliations:** Department of Biological Sciences, Wayne State University, Detroit, MI 48202, USA; ga2861@wayne.edu

**Keywords:** ubiquitin code, virus infection, virus-host interaction

## Abstract

Ubiquitination is a prominent posttranslational modification, in which the ubiquitin moiety is covalently attached to a target protein to influence protein stability, interaction partner and biological function. All seven lysine residues of ubiquitin, along with the N-terminal methionine, can each serve as a substrate for further ubiquitination, which effectuates a diverse combination of mono- or poly-ubiquitinated proteins with linear or branched ubiquitin chains. The intricately composed ubiquitin codes are then recognized by a large variety of ubiquitin binding domain (UBD)-containing proteins to participate in the regulation of various pathways to modulate the cell behavior. Viruses, as obligate parasites, involve many aspects of the cell pathways to overcome host defenses and subjugate cellular machineries. In the virus-host interactions, both the virus and the host tap into the rich source of versatile ubiquitination code in order to compete, combat, and co-evolve. Here, we review the recent literature to discuss the role of ubiquitin system as the infection progresses in virus life cycle and the importance of ubiquitin specificity in the regulation of virus-host relation.

## 1. Introduction

Ubiquitination is a reversible post-translational modification (PTM) that regulates protein functions in almost every aspect of a cell’s life. Conjugation of a 76-amino acid small protein, ubiquitin (Ub), to a target protein changes the stability, quantity and activity of that target. Ubiquitination was first discovered as a type of histone H2A modification about 43 years ago [1,2]. Later, it became clear that protein ubiquitination triggers selective degradation of the target and plays essential roles in many cellular processes [3]. After decades of efforts to delineate the regulation of ubiquitin enzymes and the network of ubiquitin linkage, now we know that the complex ubiquitin code is a major form of cellular communication that conveys distinct signals in cell pathways such as receptor signaling transport, DNA damage response, cell cycle progression, and stress responses (for recent reviews, see references [4,5,6]). 

Viruses are obligate parasites that closely interact with the host to subjugate many cellular machineries to achieve viral replication. Naturally, they have evolved to manipulate and take advantage of the ubiquitin system so as to redirect the cellular pathways in their own favor. Host cells also adopt special code of ubiquitination to mount immune responses against viral infection. It is fascinating to witness the unveiling of ubiquitin regulation in cell anti-viral defenses as well as viral counteractions in recent years. In this review we will focus on the current advances of understanding the specificity of ubiquitination in the constant battles between the virus and its host. Through delineating the complex ubiquitin code on both the virus and host factors, we hope to understand the significance of ubiquitination specificity in virus-host interaction and to search for potential targets useful for prophylactic and therapeutic treatment of viral diseases. 

## 2. Enzymes in the Ubiquitin System

Ubiquitination is carried out by a cascade of enzymatic reactions. E1 activating enzyme uses ATP to activate the C-terminal carboxyl group of ubiquitin, which then forms a thioester bond with the cysteine residue in the active site of an E2 ubiquitin conjugating enzyme. In the final step, an E3 ubiquitin ligase transfers the ubiquitin from the E2-Ub to a specific substrate. Resulted from the serial reactions, an isopeptide bond is formed between the carboxyl group of ubiquitin and the ε-amino group of a lysine residue of the substrate protein [3].

The human genome has two E1 activating enzymes, about 40 E2 conjugating enzymes and over 600 E3 ubiquitin ligases [7]. E1 serves to distinguish ubiquitin from other ubiquitin-like proteins [8]. E2s are small proteins of about 150 amino acids, whose selectivity is generally realized via their interactions with different E3s. The diverse array of E3s are categorized into three classes based on their (i) catalytic domain and (ii) the mechanism of ubiquitination reaction. The most common type of E3s is the really interesting new gene (RING) finger containing E3 ubiquitin ligases that can simultaneously bind to both the substrate and E2-Ub and directly transfer ubiquitin from the E2 to the substrate. The HECT (homologous to the E6AP carboxyl terminus) domain containing E3s and RING-between-RING (RBR) type E3s both transfer ubiquitin from E2 to form E3-Ub and then to the substrate in a two-step fashion [7,9].

Deubiquitinases (DUBs) are enzymes that remove ubiquitin from the substrate proteins. There are about 100 DUBs in humans that are classified into seven families based on their evolutionary conservation. Six of the DUB families are cysteine proteases and the last one is Zn-dependent metalloproteases [10,11].

## 3. Biochemical Diversity of Ubiquitin Code

The combination of E3 and DUB actions creates a sophisticated ubiquitin code [12], for which a meticulously balanced tuning is the key. This gives the cell enormous capacity to differentiate special signals and regulate the protein abundance, distribution and function in cell activities. Incidental errors in ubiquitination or deubiquitination can have dramatic physiological consequences leading to serious diseases such as cancer, inflammatory disease and neurodegeneration [13,14].

The specificity of ubiquitination is mainly controlled by: (i) substrate selection, (ii) lysine prioritization in the substrate, and (iii) lysine linkage in polyubiquitin chain. Selecting a substrate from the whole cell proteome and then prioritizing the lysine residues of the substrate are generally achieved through the different combinations of E2 and E3 in the network of specific E3 scaffold complex. The decision of one or multiple lysine residues to be used for monoubiquitination diversifies the possible outcome of this substrate. Moreover, all seven lysine residues (K6, K11, K27, K29, K33, K48 and K63) of the ubiquitin attached to the substrate can receive additional ubiquitin in multiple rounds of reactions. This leads to the formation of a variety of polyubiquitin chain on the substrate, including homotypic chain (only one particular lysine residue is used), heterotypic chain (different lysine residues are used), or branched chain that involves different numbers of lysine residues in each round of reaction. In addition, the N-terminal methionine residue (M1) of ubiquitin can also serve as the substrate to receive ubiquitin, which forms a unique head-to-tail type of linear ubiquitin chain [15,16,17]. To add more complexity in the ubiquitin code, non-lysine residues of the substrate protein, such as cysteine, serine and threonine can also be used in ubiquitination [18], and ubiquitin itself undergoes PTM to further increase the coding diversity of ubiquitination [4]. 

Regulatory signals embedded in the different architecture of ubiquitin are decoded by the UBD (ubiquitin binding domain) containing ubiquitin binding proteins. There are more than 20 families of UBD containing proteins in human proteome that recognize ubiquitin topologies and convey the signals to different cellular functions, the mechanisms of which are reviewed in reference [19]. 

## 4. Functional Specificity and Complexity of Ubiquitin Code

In recent years, a growing number of specific ubiquitin linkages have been allocated to particular cellular functions, demystifying the sophisticated ubiquitin language in the regulation of cell activities. The K48-linked ubiquitin chain is the most common signal that triggers the proteasome- dependent degradation of the substrates, conserved from yeast to humans [20,21,22]. However, the second most abundant form of ubiquitin chain, K63-linked chain, has little access to proteasomes [23], but is known for non-proteolytic signaling in DNA damage response, cell trafficking, autophagy and immune responses [12]. Interestingly, recent data showed that K63 ubiquitination can serve as the seed for the K48/K63 branched chain, which is then prone to target the substrate for proteasomal degradation [24], implying for more functional diversity form the complex combination of ubiquitin code.

Both K48 and K63 also form unanchored polyubiquitin chains important for intrinsic/innate immune reactions. While the K48 chains unattached to any target protein are known to activate IKKε (inhibitor of nuclear factor kappa-B kinase subunit epsilon) to promote STAT1 phosphorylation and the subsequent type I Interferon (IFN)/ISG expression [25], unanchored K63 chains are more versatile. For example, free K63 polyubiquitin chain interacts with RIG-I and activates IRF3 and NF-κB in a cell free assay, whereas in vivo it was found to activate TAK1 and upregulate NF-κB upon viral infection [26,27]. Beyond the IFN pathway, K63 chain detached from the substrate by a deubiquitinase at the proteasome can stimulate autophagy-dependent aggresome clearance, which is a critical cellular event to remove aberrant protein aggregates [28]. 

Mass spectrometry analysis has also captured a few other ubiquitin linkages, such as K11 and K29, whose levels are elevated in the presence of proteasome inhibitor, suggesting their involvement in proteasome degradation [21]. Recent data, however, indicated that the K11 chain may have dual roles in modulating protein stability under different circumstances. On one hand, K11 chain assembled by the cullin-containing anaphase-promoting complex (APC/C) marks the substrates for proteasomal degradation to control the cell cycle [29,30]. On the other hand, K11 chain attached to β-catenin leads to protein stabilization, not degradation [31]. In both cases, the same UBE2S is used as E2. Whether it is the difference in cell type, E3 enzyme type or substrate type that causes such opposite effects will need further investigation.

Due to the unevenness of linkage abundancy and the lack of linkage specific antibodies, much of the atypical ubiquitin signals, including K6, K27, K29 and K33, is understudied. Recent advancement of affimer technology, in which high affinity affimers binding to specific ubiquitin linkages can be screened from a library of randomized peptides, proves to be a valuable tool in deciphering ubiquitin code [32]. Along with the conventional mutagenesis assay and available antibodies, more details of special ubiquitin linkage and enzymes that control the linkage formation are being revealed every day. For example, it is now known that the low abundance K6 chain can be assembled by HUWE1, a HECT E3 ubiquitin ligase, or PARKIN, a RBR ubiquitin ligase, to be involved in DNA damage response or mitophagy, respectively [32,33]. It would be more interesting to find out whether these two events are correlated in the same cell and how they are coordinated.

The ubiquitin M1 linkage was first discovered when a mutant ubiquitin lacking all seven lysine residues was found to form polyubiquitin chain in vitro [34]. So far, the only E3 that catalyzes the head-to-tail true peptide bond of the M1 ubiquitin chain is the linear ubiquitin chain assembly complex (LUBAC), which contains two RING finger proteins, HOIL-1L and HOIP, and an adapter protein SHARPIN [34,35,36]. Many of the LUBAC substrates, such as NEMO, RIPK1, and TRADD, are components of TNF inflammatory signaling, indicating a critical role of M1 ubiquitin chain in innate immune responses [37,38]

## 5. Ubiquitination in Every Step of Viral Infection

With the critical role of ubiquitin modification in many aspects of a cell’s life, viruses, as obligate parasites, are found to exploit the system in every imaginable manner. They can simply adopt an existing ubiquitin regulatory pathway to hitchhike, or they can hijack the ubiquitin system for selective advantage towards the virus. They can change the specificity of cellular E3s or DUBs to modulate a network of viral/cellular substrates, or they can encode viral ubiquitin-like modifiers or ubiquitinating/deubiquitinating enzymes to modify a whole new set of substrates. Here we will follow the footsteps of virus infection and summarize how viruses manipulate host ubiquitin system to progress in infection in their own “creative” ways.

### 5.1. Role of Ubiquitination in Virus Entry

Virus entry is the very first step to start an infection. At this point all a virus can rely on is its virion proteins, so to deliver the viral genome into a proper cell machinery by the few existing proteins is paramount for initiating a successful infection. Although some viruses inject the genome by direct penetration or membrane fusion, most viruses, with or without an envelope, enter a susceptible cell via endocytosis, the major cell path to uptake substances from the environment [39,40]. Membrane receptor ubiquitination often promotes receptor endocytosis, which facilitates the removal of membrane receptor to extinguish the signal transduction [41]. Viruses broadly exploit the ubiquitination regulated endocytosis for the virus entry (Figure 1).

One example of virus adopting the endocytosis pathway for entry is to use the T cell immunoglobulin and mucin (TIM) family receptors as viral receptors. TIMs are type I transmembrane proteins that function as receptors that recognize phosphatidylserine or phosphatidylethanolamine exposed on apoptotic cells and clear them by phagocytosis [42,43]. TIMs are found to facilitate the entry of a few RNA viruses in a series of recent reports [44,45,46,47]. For example, TIM-1 serves as a Dengue virus receptor to facilitate the entry via endocytosis, and ubiquitination at K338 and K346 of TIM-1 is required for an effective virus uptake [47]. Likely, these RNA viruses, with limited virion proteins, simply take advantage of the existing receptor endocytosis promoted by ubiquitination to enhance the viral entry. The specific interaction between these RNA viruses and the extra cellular IgV-like domain and mucin domain of TIM may be the determining factor for the viral selectivity. Whether TIM recognizing apoptosis-associated phospholipid has anything to do with host apoptotic effects in intrinsic defenses remains to be investigated.

Another better studied example is the involvement of Cbl in the entry process of herpesviruses. Cbl is a RING-type E3 ubiquitin ligase, which serves as an adaptor to transduce signals in the various receptor tyrosine kinase pathways. Upon signal recognition, Cbl phosphorylation triggers its association to the receptor to ubiquitinate and remove the latter from the membrane [48,49,50]. In both the entry process of herpes simplex virus (HSV-1) and Kaposi’s sarcoma herpesvirus (KSHV), Cbl is found necessary in reducing the amount of membrane receptor via endocytosis to help the internalization of the virus [51,52,53]. Although it has been reported that an E2 conjugating enzyme CIN85 associated with Cbl interacts with an HSV-1 immediate early protein ICP0 (infected cell protein 0) [54], which is a viral RING-type E3 ubiquitin ligase by itself [55], it is not yet clear whether Cbl activity is controlled by ICP0 for a selective viral enhancement. 

### 5.2. Role of Ubiquitination in Virus Uncoating

Virus uncoating is the critical step to reveal the viral genome for replication. It is frequently associated with proteolytic cleavage and proteasome/lysosome degradation for the purpose of complete or partial removal of the capsid. Ubiquitination-mediated protein degradation naturally has an irreplaceable role in the uncoating process for many viruses.

Adenovirus (AdV), with its very complex capsid but no envelope, has long been the interest of scientists studying the stepwise viral uncoating [56,57]. Recent investigations showed broad participation of the ubiquitin system in AdV disassembly and trafficking, events occurring after the virus uptake but before its docking at nuclear pore to release the genome. On one hand, the conformational change of protein VI triggers the exposure of PPxY motif, which is recognized by a HECT-type E3 ubiquitin ligase, NEDD4. Ubiquitination of pVI helps AdV to rupture endosomal membrane and escape from the autophagy degradation [58,59]. On the other hand, a RING-type E3 ligase Mib1 is newly identified to be necessary for AdV uncoating. It is not yet clear what Mib1 ubiquitinates in this case, but most likely it is related to the microtubule interaction that moves the partially disassembled AdV towards the nucleus [60]. Unlike AdV that avoids being taken up by autophagosomes on its way to uncoat the genome, Influenza A virus (IAV) actually approaches an aggresome-autophagy pathway to help release its genome [61]. HDAC6 dependent aggresome-autophagy pathway uses HDAC6 to bridge the interactions between microtubules and various ubiquitin enzymes during the aggresome and autophagosome formation. With the recruitedDUB, unanchored K63 polyubiquitin chain is generated to stimulate the degradation of misfolded proteins [62]. Unanchored ubiquitin chains are also found inside the IAV virions. Upon entry, these polyubiquitin chains interact with HDAC6-dependent aggresome degradation machinery. Via microtubule movement, RNA segments of IAV are transported towards the nucleus and released into the nuclear pore [61]. In these two cases, both AdV and IAV take the strategy of mimicry and utilize the existing cellular pathways (Figure 1). Evidently, the specificity in choosing a particular pathway to uncoat relies on the structure and composition of the virion, which at this point of infection still is the only tool the virus can use.

Unlike other virus families, herpesviruses all contain a proteinaceous layer of tegument in the virions, which gives this family of viruses more capability in the manipulation of cell ubiquitin system in the early infection. A DUB activity has been identified in the N-terminus of the largest HSV-1 tegument protein, the 3146 amino acid protein UL36, which is conserved in all herpesviruses [63]. The C-terminus of UL36 has multiple sites to interact with capsid proteins. Therefore partial deletion of UL36 impairs its incorporation into the progeny virions, making them defective in nucleus targeting and genome release in the next round of infection [64]. Due to the difficulties in biochemical and genetic manipulation of such big protein, how UL36 affects viral uncoating is still unclear. Likely it involves a complex interaction with proteasome because proteasomal inhibition also blocks the capsid transport and genome release [65].

### 5.3. Role of Ubiquitination in Virus Replication

Once the genome is uncoated, viral expression and replication start immediately, which produces pathogen associated molecular patterns (PAMPs) that can be recognized by the host. In an attempt to suppress the virus, the host cell instantly mobilizes and synthesizes defensive molecules to mount cellular intrinsic and innate immune responses. Many well-characterized ubiquitination regulations are associated with the IFN signaling pathways (Figure 2), including the aforementioned K48, K63 and M1 linkages.

IFN was initially discovered due to its broad interference in viral infection [66,67], such as involving RNA degradation and translation shutdown to inhibit viral replication [68]. Various pathogen or damage associated molecular patterns can trigger the production of IFN, and other inflammatory cytokines, through a wide range of specific receptors and effectors and the diverse ubiquitin code [69,70]. Virus related molecular patterns generally associate with the DNA sensor- or RNA sensor-mediated IFN production, as discussed in the following subsections. 

#### 5.3.1. Ubiquitin Specificity in DNA Induced IFN Production and Viral Counteractions

Cyclic GMP-AMP synthase (cGAS) is a cytosolic protein sensing the endogenous damaged DNA or exogenous invading DNA to produce cGAMP [71], which in turn binds to stimulator of interferon genes (STING) to activate STING and transduce the signal to TANK-binding kinase 1 (TBK1) so as to phosphorylate the transcription factor IRF3. The phosphorylated IRF3 dimerizes and translocates into the nucleus to activate the transcription of type I IFN [70].

Each component in this pathway is under extensive regulation by the ubiquitin system, involving the tripartite motif (TRIM) family of RING-type E3 ubiquitin ligases [72] and some other E3s. For example, TRIM56 monoubiquitinates cGAS at K335 to enhance its binding affinity to DNA and upregulates the production of type I IFN. This ubiquitin regulation specifically restricts the infection of DNA virus HSV-1 but not the RNA virus IAV [73]. cGAS is also found to interact with RNF185 in HSV-1 infection and can be polyubiquitinated with the K27 linkage by RNF185 [74]. Whether there is a regulatory switch in the two cGAS ubiquitination reactions during HSV-1 infection is not known, neither is how the two different reactions may affect the HSV-1 pathogenesis.

The downstream effector STING has shown a tremendously complex ubiquitin regulation. For example, TRIM56, TRIM32 and mitochondrial E3 ligase MUL1 can all add K63 polyubiquitin to STING, but at different lysine residues, to stimulate TBK1-mediated IFN expression [75,76,77]. Meanwhile, an E3 ubiquitin ligase complex of autocrine motility factor receptor (AMFR) is regulated by insulin-induced gene 1 (INSIG1) to add K27 polyubiquitin to STING to stimulate the TBK1 [78]. However, RNF5 can add K48 polyubiquitin to K150 to trigger STING degradation [79], whereas RNF26 adding K11 polyubiquitin at K150 can prevent the formation of K48 chain to protect STING integrity [80]. Based on such complexity in the STING ubiquitin code, host cells may have the ability to distinguish the various forms of viral DNA with great specificity. Indeed, evidence of specific virus counteractions against STING has already been reported from human T lymphotropic virus 1 (HTLV-1) and hepatitis B virus (HBV), both of which are reverse transcribing viruses, with the former having RNA genome and latter having DNA genome. They produce different viral proteins, Tax for HTLV and polymerase for HBV, to reduce the K63 polyubiquitin chain and block the IFN production [81,82]. Whether the various ubiquitin linkages of STING have additional roles in viral replication needs further studies.

#### 5.3.2. Ubiquitin Specificity in RNA Induced IFN Production and Viral Counteractions

dsRNA is a characteristic feature of virus infection very distinguishable from the uninfected cells. Retinoic acid-inducible gene I (RIG-I) is an RNA helicase that contains two caspase recruitment domains (CARD) [83]. The RIG-I-dsRNA complex interacts the mitochondrial antiviral signaling proteins (MAVS) at the mitochondria membrane, which in turn induces the MAVS interaction to IKKε and TBK1. As a result, IRF3, IRF7 and NFκB get phosphorylated and translocated to the nucleus where they activate IFN expression [70].

TRIM25 is the first E3 ubiquitin ligase identified that catalyzes K63 polyubiquitination at the CARD domains of RIG-I to stabilize the RIG-I dimerization and dsRNA binding, which play a critical role in the RNA-sensor mediated IFN production [84]. Subsequent studies show that other E3s such as Riplet, MEX3C and TRIM4 also add K63-linked polyubiquitin to RIG-I [85]. These results indicate the E3 redundancy and the importance of regulating RIG-I K63 polyubiquitination. Naturally, in order to evade the RIG-I induced IFN production to promote virus replication, many RNA viruses have evolved specific means to inhibit the RIG-I pathway. For example, although IAV, West Nile Virus (WNV) and respiratory syncytial virus (RSV) belong to different RNA virus families, they all express their own NS1 protein to interact with TRIM25 and inhibit the K63-linked ubiquitination on RIG-I [86,87,88]. Other RNA viruses can directly code for ubiquitin enzymes to interfere the RIG-I mediated IFN production. For example, severe acute respiratory syndrome coronavirus (SARS-CoV) expresses a papain-like viral protease to mimic DUB and remove the polyubiquitin chain from RIG-I, while Toscana virus (TOSV) expresses a RBR-type E3 ubiquitin ligase to add K48-linked ubiquitin chain to RIG-I and target it for proteasomal degradation [89,90]. Besides RNA viruses, DNA viruses also generate dsRNA during infection due to the overlapped coding sequences located in both DNA strands. Therefore DNA viruses such as Epstein Barr virus (EBV) and human papillomavirus (HPV) are also found to use their specific viral proteins, BPLF1 and E6, respectively, to manipulate TRIM25 and block this critical event [91,92].

For the downstream MAVS protein, it is recently found that TRIM31 can add K63-linked ubiquitin chain to MAVS and TRIM21 can add the K27-linked chain, whereas YOD1 serves as a DUB to remove the K63 chain [93,94,95]. It is not yet clear how viruses may be modulating the MAVS ubiquitination in the infection process. A recent report showed that Zika virus (ZIKV) NS3 protein can interact with MAVS and trigger K48-linked polyubiquitination of MAVS to target it for degradation. Whether a cellular or viral E3 is responsible for the process remains to be investigated [96].

#### 5.3.3. Ubiquitin Specificity of Transcription Factors Promoting IFN Production and Viral Counteractions

The promoter of type I IFN are activated by a few transcription factors, such as IRF3 and NFκB. IRF3 can be activated by both the cGAS-STING and RIG-I pathways. Examples from both DNA and RNA viruses have been reported, in which viral proteins manipulate cellular E3s to add K48-linked polyubiquitin to IRF3 for degradation [97,98]. It is a common viral counteraction to target IRF3 and prevent IFN production. 

NF-κB is a family of key transcription factors regulating the production of IFN and many other cytokines [99]. In IFN production, upstream effectors such as RIG-I activate the IKK complex to phosphorylate IκB, which leads to the K48-linked ubiquitination and degradation of IκB. Subsequently, NF-κB, a transcription factor for multiple innate pathways, is released and translocated to the nucleus to activate the expression of IFN, along with other cytokines [100]. Since IκB degradation is the key step in NFκB activation, both DNA and RNA viruses frequently target this step to control the NFκB activity. Some viruses express proteins, such as NSP1 of rotavirus and ORF61 of Varicella-Zoster Virus, to directly interact with the E3 ubiquitin ligase and block the IκB ubiquitination, [101,102]. Other viruses, such as HSV-1, use viral DUB to directly deubiquitinate IκB to prevent its degradation [103]. Interestingly, not all viruses inhibit IκB degradation. HTLV-1 Tax has been reported to interact with LUBAC to add M1 ubiquitin and generate the M1/K63 hybrid linkage on IKK, which constantly activates IKK and prevents NF-κB from being inhibited by IκB [104]. As aforementioned, Tax also modulates STING ubiquitination to counteract IFN production [81], the seemingly contradictory reports may reflect a complex coordination between IFN and inflammatory cytokine regulation in the HTLV-1 infection.

#### 5.3.4. Ubiquitin in IFN Responses and Viral Counteractions

IFN works in both autocrine and paracrine manners to amplify the anti-viral effects by stimulating the expression of many interferon stimulated genes (ISGs). In IFN responsive pathways, individual IFN family members are recognized by specific membrane receptors to initiate signaling transduction, by which the IFN-receptor complexes activate members of the Janus kinase (JAK) family to phosphorylate the signal transducers and activators of transcription (STAT) family of transcription factors, which in turn promote the transcription of ISGs [105,106]. 

To evade the massive interference of IFN, viruses have also evolved ways to inhibit the IFN responses, in which ubiquitination again plays an important role. On the IFN receptor level, hemagglutinin (HA) protein of IAV is found to induce a K48-linked ubiquitination of IFNα receptor 1 (IFNAR1) and downregulate its membrane abundancy [107]. IFNAR1 ubiquitination is also observed in HBV infection, in which a cellular protein matrix metalloproteinase 9 (MMP9) is activated to mediate the IFNAR1 ubiquitination in order to reduce its membrane abundancy [108]. 

Because of the extensive cross-talking of JAK/STAT signaling in the overall cytokine responses, the details of viral specific regulation via the JAK and STAT proteins are not clearly understood. For example, NS1 protein of RSV, NS5 protein of Dengue virus and Zika virus, and V protein of mumps virus are all found to trigger proteasomal degradation of STATs. Some of the viral proteins interact with components in E3 ubiquitin ligase complexes, while others may act as E3 ubiquitin ligases by themselves [109,110,111,112]. At this point, details in the viral specific regulation are still missing. Further investigations are needed. 

#### 5.3.5. Ubiquitin Specificity in Intrinsic Responses and Viral Counteractions

To create an environment that supports viral transcription and replication, viruses heavily manipulate cell cycle and cause cell stress, which frequently trigger the cell level intrinsic anti-viral responses, involving pathways such as apoptosis, chromatin remodeling and stress responses. 

In virus infection, both the host and virus attempt to manipulate the apoptotic pathways to their own advantage. From the host, apoptosis helps to shut down cell machineries to constrain the infection, whereas the virus exploits apoptosis for selective viral expression and virus dissemination [113]. In this process, virus also manipulates the ubiquitin system to achieve a fine balance between the proapoptotic and antiapoptotic activities. For example, tumor suppressor p53 plays a pivotal role in DNA damage response and apoptosis. 

Many viruses have evolved various strategies to up- or down-regulate the p53 activity to capitalize on the pro/anti-apoptotic molecules according to their needs. For example, the E6 protein of HPV is famous for hijacking a cellular E3 ubiquitin ligase E6AP to degrade p53 by polyubiquitination [114]. Recent results showed that R175 residue of p53 is essential for the p53-E6 interaction which bridges for an E6/E6AP/p53 complex. Interestingly, E6 itself is also ubiquitinated by E6AP, so decrease in E6 stability or blockage in the p53-E6 interaction upregulate the p53 level [115]. It is plausible to hypothesize that the fine tuning of the trimeric interactions to achieve a balance in E6 and p53 levels may be associated with the persistent HPV infection in differentiating epithelium, the cause of HPV induced cancer. Caspase family proteins are also important modulators for apoptosis. A20 is an E3 ubiquitin ligase that catalyzes the K63-linked polyubiquitination of caspase 8. In HBV infection, HBx protein promotes apoptosis in the infected hepatocytes by inducing the expression of miR-125a, a microRNA for A20 [116]. Whether a balanced ubiquitination of caspase 8 plays a role in HBV tumorigenesis has not yet been investigated (Figure 3).

Nuclear domain 10 (ND10) is a dynamic nuclear structure that has promiscuous functions in many cellular events such as DNA repair, gene regulation, cell cycle regulation, and antiviral defenses. This nuclear structure has over 150 components assembled together through the interaction of SUMOylated proteins [117]. ND10 is frequently found to colocalize at the incoming viral DNA, and some of its major component proteins restrict viral transcription and replication via chromatin repression [118]. Many viruses express specific viral proteins to disrupt or modify ND10 structure while establishing the replication, such as ICP0 of HSV-1, pp71 of human cytomegalovirus (HCMV), and E4-ORF3 of AdV [119]. Among them, ICP0 is the best studied viral RING-type E3 ubiquitin ligase that ubiquitinates the organizer of ND10, promyelocytic leukemia (PML) protein, and targets it for proteasomal degradation (Figure 3) [120]. The unique phenomenon in this ubiquitin regulation is that ICP0 has the ability to distinguish the different isoforms of PML, which share the same N-terminal exons 1-6 and differ only at their C-terminus [121,122]. Whether the specific recognition of these similar substrates by ICP0 can lead to distinct ubiquitin linkages or differential roles in viral infection remains unclear. RTA protein of KSHV also has E3 ubiquitin ligase activity to target the SUMOylated PML for degradation [123]. It is not yet clear whether the ability of ICP0 to differentially recognize similar substrates is conserved in other members of the herpesvirus family. 

APOBEC (apolipoprotein B mRNA editing enzyme catalytic polypeptide) is a family of cytosine deaminases that are capable of the C to U RNA editing. Therefore, they act as a layer of intrinsic immunity to restrict the replication of retroviruses and some DNA viruses through the induction of hyper mutations [124]. A classic example of viral counteraction of APOBEC is the Vif protein of HIV-1, which induces the degradation of APOBEC3G mediated by the Cul5-SCF E3 ubiquitin ligase [125]. In the absence of Vif, APOBEC3G is packaged into the virions, which induces hyper mutations in the cDNA synthesis upon infection and therefore impair viral genome integrity and virus production (Figure 3) [125,126]. The same virus-host interaction also exists in the reverse transcribing HBV infection, in which viral protein HBx upregulates the MSL2 E3 ubiquitin ligase to trigger the APOBEC3B degradation [127]. 

#### 5.3.6. Ubiquitin Specificity in Viral Replication Enzymes

Besides relying on host transcription/translation machineries, many viruses encode their own DNA/RNA polymerases to carry out viral transcription and genome replication. Ubiquitin regulation on these viral encoded replication enzymes has also been reported. For example, all three subunits (PA, PB1 and PB2) of the IAV RNA polymerase are ubiquitinated to enhance the genome replication [128]. More detailed study now have revealed that Cul4 E3 ubiquitin ligase can add K29-linked polyubiquitin to PB2 to improve replication [129]. The significance of the ubiquitination on other subunits and how these ubiquitin codes coordinate to regulate the IAV genome replication remain to be seen. Similarly, Vp35 of Ebola virus (EBOV) is a cofactor for its RNA polymerase. In EBOV infection, Vp5 interacts with the cellular E3 ubiquitin ligase TRIM6 for itself to be ubiquitinated in order to enhance the genome replication [130]. 

Ubiquitination is also important in modulating the host DNA-dependent RNA polymerase in viral transcription. The Tat protein of HIV is well known for substantially improving HIV gene transcription [131]. Recent results revealed that Tat does so by hijacking a cellular E3 ligase UBE2O to monoubiquitinate HEXIM1, an inhibitory protein that binds to the positive transcription elongation factor b (P-TEFb). The non-degradative monoubiquitination of HEXIM1 leads to the release of P-TEFb and drives RNA pol II into the elongation mode [132]. Presumably, Tat may promote the transcription elongation in a viral specific manner to tilt the reaction towards HIV RNA production, but evidence still remains to be seen. 

### 5.4. Role of Ubiquitination in Virus Egress

Virus assembly and release is the last but not least step in a virus infection cycle. Without this step, a continuous viral propagation via shedding and transmission cannot be ensured. Most of the molecular foundation for viral egress comes from studying the budding process of enveloped viruses, a process depending on the endosomal sorting complex required for transport (ESCRT). Similar to the role of ubiquitination in the receptor internalization during the virus entry, ubiquitination of cargo membrane proteins that marks them for ESCRT recognition is again important for the viral egress [133]. 

Several families of enveloped viruses contain late assembly domain (L domain) in their Gag proteins, including HIV, EBOV and RSV. One conserved feature for these viruses is the presence of PPxY motif in the L domain that can be recognized by the NEDD4 family of E3s for K63-linked ubiquitination, a key to the efficient budding of these viruses [134,135,136]. In HIV-1 variants that lack the PPxY motif, the virus can circumvent the defect by using a cellular protein, angiomotin, to bridge the interaction of Gag to a NEDD4-2 isoform [137,138]. The extensive efforts in Gag ubiquitination indicate its importance in the HIV egress, likely to compensate for the high mutation rate and slow productivity of HIV. 

## 6. Conclusions

Virus replication relies on its abilities to overpower the host defenses as well as to harness the cell machinery. To achieve both goals simultaneously, it is essential for the virus to walk a fine line in sorting out friends and foes in the host cell proteome. Ubiquitin modification has enormous power to modulate the protein stability, specificity, and affinity to fine tune its position in an interaction network. Therefore the ubiquitin system is an ideal cell machinery for the virus to adopt and manipulate to its own advantage. Delineation of virus specificity in controlling the ubiquitin system to write viral ubiquitin code will help us to identify the beneficial and inimical factors in a virus infection. This will become the cornerstone in developing novel treatment strategies for viral diseases.

## Figures and Tables

**Figure 1 ijms-21-04088-f001:**
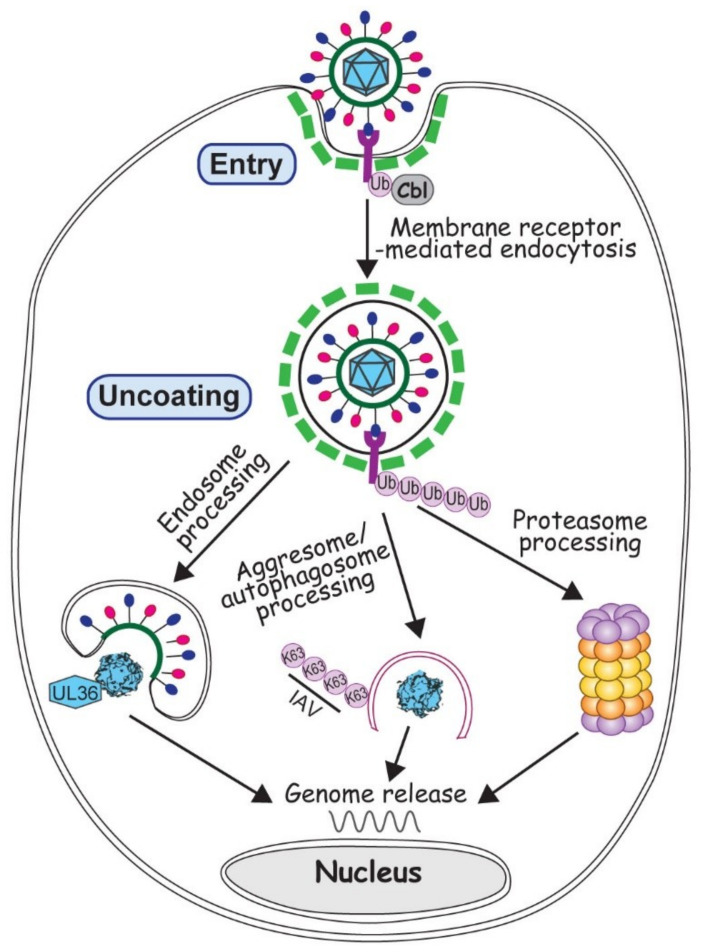
Involvement of ubiquitin in virus entry and uncoating. Virus entry is depicted by a generic virus (with red and blue ellipsoids attached to a green circle representing viral envelope and glycoproteins, and cyan polygon representing viral neucleocapsid) interacting with the membrane receptor colored in magenta. Ubiquitination of membrane receptor promotes virus entry via endocytosis (with green dashed lines representing the clathrin coat). Upon entry, viruses can employ endosome, autophagosome or proteasome to process the virion and release the viral genome, as described in the text.

**Figure 2 ijms-21-04088-f002:**
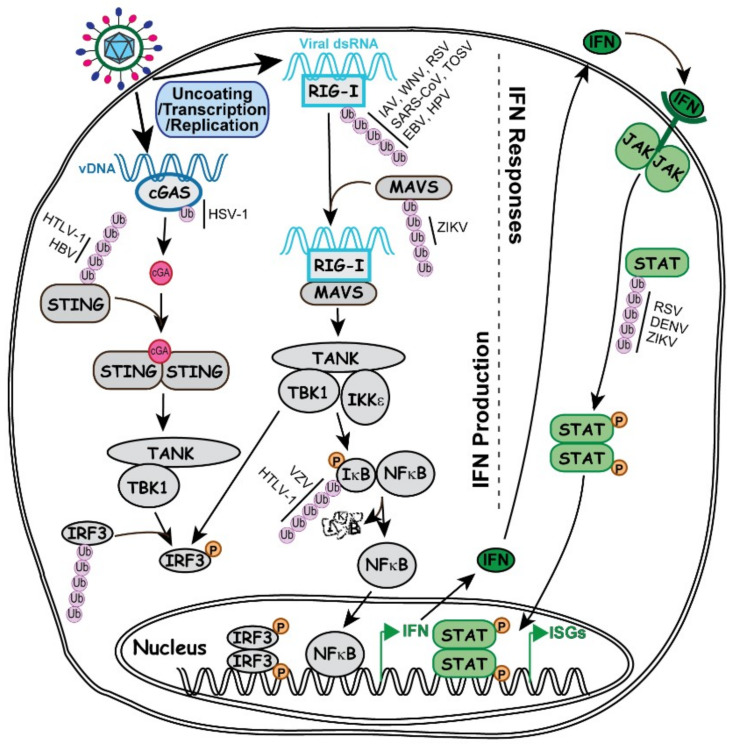
Involvement of ubiquitin in the key signaling pathways important for IFN production and IFN responses. Illustrated on the left side of the grey dashed line, abnormal presence of vDNA (in dark blue) or viral dsRNA (in light blue) is detected by DNA sensor cGAS (Cyclic GMP-AMP synthase) or RNA sensor RIG-I (Retinoic acid-inducible gene I), which triggers the activation of their respective signaling to induce IFN production as described in the text. On the right side of the grey dashed line, secreted IFN interacts with IFN receptor depicted in green Y-shaped line. IFN and receptor work in either autocrine or paracrine fashion to induce the activation of JAK/STAT pathway and promote ISG expression. Steps of IFN production and responses involving ubiquitination and viral regulations of the ubiquitination process are discussed in the text.

**Figure 3 ijms-21-04088-f003:**
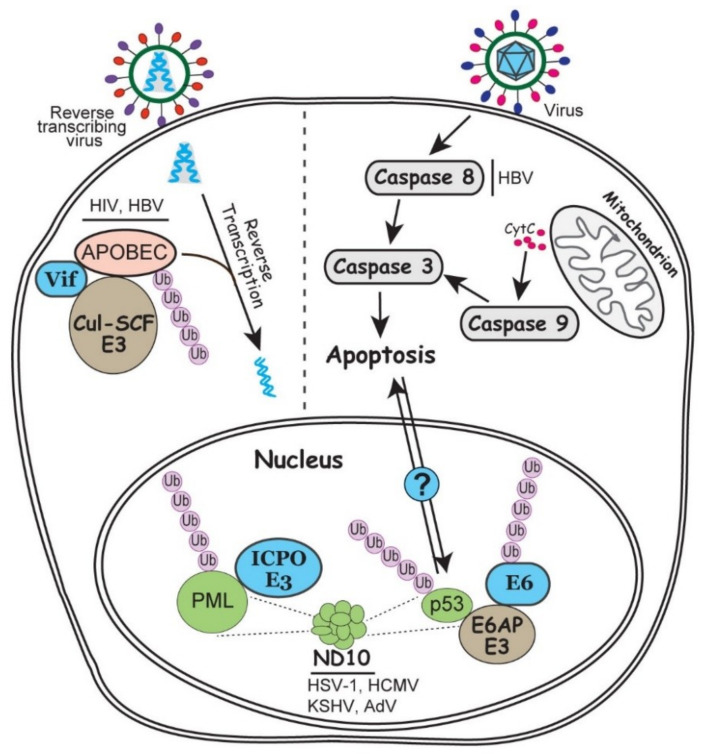
Involvement of ubiquitin in the intrinsic antiviral responses including apoptosis, ND10 nuclear bodies and APOBEC enzymes. In the nucleus, ND10 associated PML and p53 regulations are enlarged to show detailed interactions. On the right side of the grey dashed line, infection by a generic virus (as described in Figure 1) can trigger exogenous or endogenous apoptosis via caspase activation. Potential communication of the p53 and caspase induced apoptosis is indicated by a question mark. On the left side of the grey dashed line, a reverse transcribing virus associated with APOBEC editing is illustrated. Steps involving ubiquitination and viral regulation of the ubiquitination process are discussed in the text.

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
