# Peer review of "Specificity in Ubiquitination Triggered by Virus Infection"

_ijms, 2020, doi:10.3390/ijms21114088_

Round 1
Reviewer 1 Report
This paper summarizes recent publications on the roles and regulation of ubiquitin code in virus infection. It is written in a concise and clear manner. This Reviewer asks the authors to add schematic diagram of ubiquitin-mediated infection pathways to make readers to understand more easily.
Author Response
This paper summarizes recent publications on the roles and regulation of ubiquitin code in virus infection. It is written in a concise and clear manner. This Reviewer asks the authors to add schematic diagram of ubiquitin-mediated infection pathways to make readers to understand more easily.
We thank Reviewer #1 for the kind encouragement towards our manuscript. We have included three schematic figures in the revised manuscript.
Reviewer 2 Report
Gu and Fada describe how viruses exploit cellular ubiquitination machinery and ubiquitin code for their successful infection. The authors summarize viral intervention on cellular ubiquitination machinery along every step of the infection pathway. Largest portion is assigned to description of role of ubiquitination in viral replication, with particular emphasis on intervention of interferon signaling upon viral infection.
Given the recent global pandemic by viral infections and apparently small number of relevant extensive reviews in the literature, the manuscript would in principle give insights to broad audience in the related fields. However, the reviewer feels that the manuscript in current form may not be suitable for publication as it is and that more focusing in terms of the contents with emphasis on mechanistic insights and placement of figure(s) would be needed.
Major concerns:
1. The reviewer strongly recommend the authors to focus more on the current section 5.3 with everything else being either deleted or minimized. Also suggested is the consideration of changing the title; one suggestion would be "How viruses intercept cellular ubiquitination machinery for their replication" or something like that in order to narrow the scope of the review for the benefits of focusing.
2. Sections 1 through 4 in the current manuscript need to be shortened so that only essential information would be introduced.
3. The authors mainly list viral proteins that have been reported to interfere with cellular ubiquitination machinery in viral infection without sufficient mechanistic insights (for example, lines 267-269; 305-308). If the authors restrict the scope of the review to ubiquitination in viral replication and focus on description of mechanistic aspects of the roles of viral proteins in interfering with cellular ubiquitination machinery, the review would appeal more attractively to the readers in broader areas.
4. The authors are strongly encouraged to place at least one figure summarizing how viral proteins intercept ubiquitination machinery. A great figure would be worth 100 pages of well-written text.
Minor points:
Some sentences tend to be long; it would be better read if the authors make sentences little bit shorter: for instance, lines 305-308.
Line 357: uqbituinated -> ubiquitinated
Line 421: k63 -> K63
Author Response
Gu and Fada describe how viruses exploit cellular ubiquitination machinery and ubiquitin code for their successful infection. The authors summarize viral intervention on cellular ubiquitination machinery along every step of the infection pathway. Largest portion is assigned to description of role of ubiquitination in viral replication, with particular emphasis on intervention of interferon signaling upon viral infection.
Given the recent global pandemic by viral infections and apparently small number of relevant extensive reviews in the literature, the manuscript would in principle give insights to broad audience in the related fields. However, the reviewer feels that the manuscript in current form may not be suitable for publication as it is and that more focusing in terms of the contents with emphasis on mechanistic insights and placement of figure(s) would be needed.
Major concerns:
- The reviewer strongly recommend the authors to focus more on the current section 5.3 with everything else being either deleted or minimized. Also suggested is the consideration of changing the title; one suggestion would be "How viruses intercept cellular ubiquitination machinery for their replication" or something like that in order to narrow the scope of the review for the benefits of focusing.
Many of the ubiquitin regulated signaling pathways delineated so far are related to the complex IFN signaling or TRIM family of E3s. Therefore there are many reviews in the field discussing the topic of ubiquitin regulated innate immunity. We have referred quite a few in our manuscript for the background introduction. However, we did not want to simply repeat the immunologists’ viewpoints in our manuscript. We intended to summarize the ubiquitin regulation in the view of viral replication, which include the essential steps of virus entry, uncoating, replication, assembly and egress. Although some parts of the field are still young compared to the IFN signaling, specific viral regulation on the ubiquitin system is involved in all these steps. In our manuscript, we have reviewed the status for all infection steps, and pointed out some of the directions to pursue for more investigations. We do not think the reviewer’s suggestion reflects what we want to achieve for this manuscript.
- Sections 1 through 4 in the current manuscript need to be shortened so that only essential information would be introduced.
We have revised these sections to make them more succinct.
- The authors mainly list viral proteins that have been reported to interfere with cellular ubiquitination machinery in viral infection without sufficient mechanistic insights (for example, lines 267-269; 305-308). If the authors restrict the scope of the review to ubiquitination in viral replication and focus on description of mechanistic aspects of the roles of viral proteins in interfering with cellular ubiquitination machinery, the review would appeal more attractively to the readers in broader areas.
As aforementioned in point #1, some parts of the virology field have just started to investigate the involvement of ubiquitination, such as uncoating and egress. Detailed mechanisms are quite lacking in these areas, especially if we consider different viruses take very diverse approaches to adopt or attack the ubiquitin system. This is exact the purpose of this manuscript – to summarize what is known and what is not, and to indicate where to go in the field of ubiquitin regulation in virology. We did not want to only include the well documented interactions between ubiquitination and innate immunity. We believe our manuscript provides a fresh look at the ubiquitin system from a virologist’s point of view, which brings unique insights and will appeal to broader interests in the fields of both immunology and virology.
- The authors are strongly encouraged to place at least one figure summarizing how viral proteins intercept ubiquitination machinery. A great figure would be worth 100 pages of well-written text.
We have included three schematic figures in the revised manuscript.
Minor points:
Some sentences tend to be long; it would be better read if the authors make sentences little bit shorter: for instance, lines 305-308.
We revised the long sentences throughout the text.
Line 357: uqbituinated -> ubiquitinated
Corrected.
Line 421: k63 -> K63
Corrected.
Round 2
Reviewer 2 Report
It seems that the authors addressed the issues raised by the reviewer satisfactorily. Although the reviewer does not necessarily agree with what the authors want to achieve in the manuscript, the authors' claim is understandable.
The figures are fine. However, figure legends need to be improved. Specifically: Figure 1 legend: Description of what the red and blue colored ellipsoids (I guess they represent viral surface proteins), and cyan polygon (genetic material with coat proteins?) are. The green (membrane fragment?) dashed lines are also need to described. All abbreviations such as UL36 and IAV should be explained in full. Figure 2 legend: IFN receptor (green Y-shaped line) needs to be specified. All abbreviations for virus names need to be fully written. Figure 3 legend: Two different types of viral genetic materials should be clearly indicated. Abbreviations for protein names need to be fully written as well. What do the authors mean by just "virus" in contrast to "HIV"?
Author Response
It seems that the authors addressed the issues raised by the reviewer satisfactorily. Although the reviewer does not necessarily agree with what the authors want to achieve in the manuscript, the authors' claim is understandable.
We thank reviewer 2 for the understanding comments.
The figures are fine. However, figure legends need to be improved. Specifically: Figure 1 legend: Description of what the red and blue colored ellipsoids (I guess they represent viral surface proteins), and cyan polygon (genetic material with coat proteins?) are. The green (membrane fragment?) dashed lines are also need to described. All abbreviations such as UL36 and IAV should be explained in full. Figure 2 legend: IFN receptor (green Y-shaped line) needs to be specified. All abbreviations for virus names need to be fully written. Figure 3 legend: Two different types of viral genetic materials should be clearly indicated. Abbreviations for protein names need to be fully written as well. What do the authors mean by just "virus" in contrast to "HIV"?
We modified figures and revised the legends (see the revised manuscript). Abbreviations in figures have been spelled out in the text.